# Nitrogen Utilization in Goats Consuming Buffelgrass Hay and Molasses-Based Blocks with Incremental Urea Levels

**DOI:** 10.3390/ani13213370

**Published:** 2023-10-30

**Authors:** Zaida Torres-Cavazos, Daniela S. Rico-Costilla, Gustavo Moreno-Degollado, Sara P. Hernández-Martínez, Gerardo Mendez-Zamora, Yareellys Ramos-Zayas, Jorge R. Kawas

**Affiliations:** 1Facultad de Medicina Veterinaria y Zootecnia, Universidad Autónoma de Nuevo León, Avenida Francisco Villa S/N, Colonia Ex-hacienda El Canadá, Escobedo 66050, Nuevo León, Mexico; z.torresc@hotmail.com (Z.T.-C.); daniela.ricocst@uanl.edu.mx (D.S.R.-C.); gustavo.morenod@uanl.mx (G.M.-D.); yramosz@uanl.edu.mx (Y.R.-Z.); 2Facultad de Agronomía, Universidad Autónoma de Nuevo León, Avenida Francisco Villa S/N, Colonia Ex-hacienda El Canadá, Escobedo 66050, Nuevo León, Mexico; sara.hernandezma@uanl.edu.mx (S.P.H.-M.); gerardo.mendezzm@uanl.edu.mx (G.M.-Z.); 3MNA de México, Avenida Acapulco 770, Colonia La Fe, San Nicolás de los Garza 66477, Nuevo León, Mexico

**Keywords:** goats, molasses–urea blocks, nitrogen utilization, protein requirements

## Abstract

**Simple Summary:**

Protein in the diet is essential for the growth and health of ruminants and other animal species. Protein is also one of the most expensive components in animals’ diets, and finding good protein sources can sometimes be challenging. Urea supplementation, as a non-protein nitrogen source, is often recommended for goats fed low-quality forages. In the rumen, microorganisms transform urea into microbial protein that can be assimilated by the animal. In this study, we investigated nitrogen utilization in goats fed low-quality hay supplemented with molasses blocks containing various levels of urea. The findings and discussion in this paper contribute to a better understanding of nitrogen utilization in goats using urea as a non-protein nitrogen source.

**Abstract:**

The use of goats for meat production faces challenges from environmental and nutritional factors. Urea is an affordable non-protein nitrogen source commonly utilized in ruminant nutrition. The objective of this study was to investigate nitrogen utilization in goats fed low-quality hay supplemented with molasses blocks containing urea. Twenty Anglo-Nubian doelings were individually housed in metabolic cages and provided with chopped Buffelgrass (*Cenchrus ciliaris*) hay ad libitum. Goats were randomly assigned to four urea levels (0, 2, 4, and 6%; *n* = 5 per treatment) in molasses blocks for a duration of 30 days. A negative nitrogen balance (−2.458 g/day) was observed in doelings consuming blocks without urea, compared with a positive balance (0.895 g/d) for those consuming the 6% urea blocks. Block nitrogen intake significantly increased with urea level, but urea supplementation did not affect dry matter (DM) or neutral detergent fiber (NDFom) intake or digestibility. A minimum crude protein (CP) requirement of 8% for maintenance in doelings consuming low-quality forage with a urea-based supplement was determined through regression analysis between CP intake (% of DM) and N balance (r^2^ = 0.479; *p* < 0.002). The value of 8% of CP obtained in this study is similar to several previous studies reported in the literature, but in this case, the increments in CP came exclusively from urea. In this study, increasing the urea content of molasses blocks up to 6% significantly increased nitrogen intake, retention, and balance in goats. These results contribute to a better understanding of nitrogen utilization in goats fed low-quality hay with urea supplementation.

## 1. Introduction

Goats (*Capra aegagrus hircus*) are small ruminants capable of producing meat, milk, and other valuable products (cashmere and mohair) for humans. Most breeds are characterized by their ability to thrive in environmentally adverse conditions [1]. The global goat population exceeds one billion and continues to grow, particularly in Asia and Africa [2]. In some countries, especially those in semiarid regions, goats play a significant role in local economies and the livelihoods of smallholders [3]. However, the potential of goats to grow, reproduce, and yield useful products is challenged by environmental factors (e.g., weather) and nutritional factors (e.g., diet composition) [4].

In certain regions, goats are primarily fed with locally available grass and crop residues, but these feeds often fall short of providing the necessary energy, protein, minerals, and vitamins. Forage availability and quality are limiting factors. This is a concern because undernourished animals cannot fully realize their full potential to maintain health, reproduce, and simultaneously produce meat and milk [5]. Nutrient supplementation for animals consuming low-quality forages can have a positive impact, but it should be specific and take into consideration the protein and mineral concentrations of forages in different regions [1,6].

Dietary crude protein and nitrogen supply to the animal are vital for maintaining health, as amino acids and nitrogen are involved in the production of essential molecules such as antibodies, enzymes, and neurotransmitters [7]. However, nitrogen metabolism is not well understood because it involves different pathways for absorption and excretion, depending on the type and quantity of the dietary protein, as well as the protein synthesis rates in different organs [8,9]. Urea is a common ingredient in ruminant nutrition as a non-protein nitrogen source and has been used for over a century [10]. Urea is metabolized by ruminal microbes, which in turn produce protein that the host can assimilate [11,12]. Molasses blocks are an intriguing alternative for supplying urea as non-protein nitrogen source because the efficacy of urea microbial degradation depends on the presence of fermentable carbohydrates [13,14]. Additional advantages of molasses-urea blocks include easier transportation and more consistent consumption among animals.

Buffelgrass (*Cenchrus ciliaris*) is the predominant warm season perennial grass in northeastern Mexico and is also common in other semi-arid regions worldwide. It is highly productive and tolerant to the periodic droughts that occur in these regions [15]. The objective of this study was to investigate the effect of supplementing molasses blocks with various levels of urea on nitrogen utilization in Anglo-Nubian female doelings fed Buffelgrass hay.

## 2. Materials and Methods

### 2.1. Animals, Supplements, and Feeding Management

This study was approved by the Joint Graduate Program of the Faculties of Agronomy and Veterinary Science of the University of Nuevo León and registered under the code 36397-001290684 in the masters’ exam certificate and code 4768 in the digital collection of the masters’ degree thesis. Twenty crossbred Anglo-Nubian doelings, aged 6 to 8 months, with average body weights of 18.2 kg, were randomly assigned to one of the four treatments in a completely randomized-designed experiment. The treatments consisted of molasses blocks containing incremental urea levels of 0, 2, 4, and 6%, which were supplemented to doelings fed Buffelgrass hay ad libitum. These levels of urea were selected to provide enough protein for ruminal activity without the risk of intoxication [16].

The animals were housed in individual metabolic cages (0.9 m × 1.2 m) equipped with water and feed troughs. Daily water intake of doelings was measured. The study spanned 30 days, with the first 21 days dedicated to adapting the doelings to metabolic cages and Buffelgrass hay, as well as block consumption. The subsequent 9 days were allocated for data collection, including body weights, feed intake, and feces and urine excretion.

The ingredients and chemical composition of the block supplements are presented in Table 1. These block supplements were manufactured using a mechanical block press, modified with a hydraulic jack [17]. The ingredients used in the formulation of the supplements included soybean hulls, cracked corn, urea, salt, calcium oxide, and a mineral premix with vitamin A. The four treatment block supplements contained increasing levels of urea, replacing cracked corn.

Animals were provided with ad libitum access to chopped Buffelgrass (*Cenchrus ciliaris*) and their respective molasses block. The chemical composition of Buffelgrass was as follows: crude protein, 6.47%; crude fat, 0.94%; NDFom, 69.2%; ADF, 48.1%; ADL, 7.79%; and ash, 9.36%. Block intake was calculated as the difference between daily morning block weights. Total hay was offered in two portions during the day (09:00 and 16:00 h). Rejected Buffelgrass hay was weighted and recorded in the morning. At the end of the experimental period, time dedicated to eating, ruminating, or engaging in other activities was recorded every 5 min over a 24 h period [18].

### 2.2. Feed Sample Collection and Analysis

Offered and rejected hay samples were frozen for further analysis. Hay and ort samples were dried in an air-draft oven at 55 °C and ground through a 1 mm screen in a Wiley Mill before analysis. Dry matter content was determined at 105 °C [19]. Ash content was determined after sample combustion in a muffle furnace at 600 °C for 3 h, and ether extract was determined using the Ankom Technology XT10 Extractor. Nitrogen content of feed was determined using the micro-Kjeldahl procedure [19]. Crude protein (CP) was calculated as N × 6.25. Neutral detergent fiber (NDF) analysis was determined using the Ankom Technology model A200 fiber analyzer with filtration bags [20], and ash-free NDF (NDFom) was calculated [21]. Metabolizable energy was calculated using values reported for ingredients by the National Research Council [22]. Non-fibrous carbohydrates (NFC) content of the rations were estimated using the formula:*NFC (%) = DM − (CP + EE + ash + NDFom).*(1)

### 2.3. Feces and Urine Collection and Analysis

Daily samples of total fresh feces were collected from each doelings and then frozen. The nine fecal samples of each doeling were further thawed and mixed into a composite sample. Composite fecal samples were dried in an air-draft oven at 55 °C and ground through a 1 mm screen in a Wiley Mill. Urine was collected and weighed daily, and a daily 10% sample was accumulated in plastic containers for 9 days and immediately frozen at a temperature below −20 °C to prevent N loss. Urine was thawed and filtered through a fiber glass layer. Nitrogen in feces and urine was determined using the micro-Kjeldahl method [19] and used to calculate nitrogen balance and nitrogen retention:*N balance (g/d) = N consumed − (N in feces + N in urine)**Retained N (%) = (N balance/N intake) (100)*(2)

### 2.4. Blood Analysis

Blood samples were obtained from each doeling at the end of the experiment (day 30). All samples were thawed for 30 min at ambient temperature and centrifuged at 1000× *g* for 15 min. Serum was separated and frozen at −72 °C until analysis to determine plasma urea nitrogen concentrations. Blood urea was determined using the Berthelot colorimetric method (Idexx Laboratories Inc., Westbrook, ME, USA).

### 2.5. Rumen Fluid Collection and pH Analysis

To obtain rumen fluid samples from the doelings, an esophageal probe was inserted orally. The rumen fluid was collected in 50 mL conical tubes, and pH was immediately measured using a Beckman pH meter.

### 2.6. Statistical Analysis

All data were analyzed using an analysis of variance for a completely randomized design using Statistics 9 Analytical Software (Tallahassee, FL, United States). The model included treatments, and all possible interactions, with the animal as experimental unit. Animal and the error term were considered random in the model. The statistical model used was as follows:*y_ij_ = µ + T_i_ + µ_ij_*(3)
where *y_ij_* is the response variable for the *i*th treatment (where *i* = 1, 2, 3, 4) and the *j*th observation treatment group (where *j* = 1, 2, …, n*_i_*); *µ* is the overall population mean (the average response across all treatments); *T_i_* is the effect of the *i*th treatment level (incremental urea level), which represents the difference between the mean response for treatment *i* and the overall population mean. The treatments (urea levels) are fixed effect which represent the main factor being study.

All variables were analyzed for lineal and quadratic responses to urea levels using orthogonal contrasts. The Tukey multiple comparison test was used to determine differences among means. Mean *p*-values were considered statistically significant at *p* < 0.05. The initial weight of the does was considered as a covariate. Correlations coefficients were obtained between block intake, water consumption, urine excretion, fecal excretion, and dry matter digestibility. A lineal regression analysis was performed to determine the maintenance crude protein requirement of adult doelings using diet crude protein and nitrogen balance data.

## 3. Results

### 3.1. Chemical Composition of Blocks

The crude protein content increased from 4.1% for blocks without urea to 21.7% for blocks containing 6% urea (Table 1). In contrast, NDFom decreased from 19.9% to 13.4% as urea replaced corn in the supplement. Blocks, with the inclusion of calcium oxide, salt, and minerals, exhibited high ash contents (ranging from 32.3 to 34.9%), and low energy density (varying from 1.933 to 2.095 Mcal ME/kg DM).

### 3.2. Dry Matter and NDFom Intake and Digestibility

No significant difference (*p* > 0.05) was observed in dry matter intake (DM) or digestibility among treatments (Table 2). Forage intake varied between 542 and 571 g/day, while block intake ranged from 121 to 168 g/day. Total DM intake varied from 663 and 736 g/day among treatments. Hay rejection was high, the percent of orts being 30.5, 34.5, 33.9, and 32.6% of Buffelgrass offered, for 0, 2, 4, and 6% of urea treatments, respectively.

Due to the high inorganic matter content of the blocks, DM digestibility was lower than NDFom digestibility. Although the NDFom content in Buffelgrass was high (69.2%), due to doelings’ dietary selection, the forage consumed contained less NDFom (58.8, 59.7, 56.2, and 59.7% for 0, 2, 4, and 6% urea treatments, respectively). The NDFom in orts was 87.0, 86.5, 85.5, and 85.4%, respectively. The Fecal NDFom was 52.9%, 50.8%, 49.4%, and 52.1%, respectively. These low fecal NDFom values suggest that forage consumed was low in NDFom content.

### 3.3. Time Dedicated to Eating and Ruminate

The time dedicated to eating was not affected (*p* > 0.05) by urea block level, ranging from 412 to 471 min/day. In contrast, rumination time showed a quadratic response (*p* = 0.013) to an increased urea level in the blocks (Table 3). The total chewing time (sum of eating and rumination times) was not affected (*p* > 0.05) by block urea level. The ruminal pH of the doelings changed quadratically (*p* = 0.012) with more urea in the blocks (Table 3).

### 3.4. Nitrogen Intake and Retention

In this study, although buffelgrass contained 6.47%, due to forage selection, the crude protein of rejected forage ranged from 4.98 to 5.19%. The nitrogen balance data from two doelings were discarded due to urine lost from two metabolic cages during the 9-day urine-collection period. Whereas forage nitrogen intake did not differ (*p* > 0.05) among the treatments, block nitrogen intake increased (*p* < 0.001) from 1.065 g/d with the block without urea to 4.6 g/d with the block with 6% urea (Table 4), resulting in an increasing total nitrogen intake from 7.180 g/day to 10.605 g/day. No significant (*p* > 0.05) increase in fecal or urine N excretions was observed with an increasing block urea inclusion level (Table 4). With more urea in blocks, nitrogen retention (*p* = 0.005) and N balance (*p* = 0.010) linearly increased. Nitrogen balances shifted from negative (−2.458 g/day) for doelings consuming blocks without urea to positive (0.895 g/d) for those consuming the 6% urea blocks. Doe nitrogen retention shifted from negative (−35.93%) for blocks without urea to positive (10.06%) for the 6% urea block. The nitrogen balances of the doelings increased with more crude protein consumed (Figure 1). Considering a zero N balance, the maintenance CP requirement of the doelings was calculated to be 8%.

### 3.5. Blood Urea Nitrogen

Blood urea concentrations linearly increased (*p* < 0.001) from 5.8 to 15.2 mg/dL as the urea inclusion in blocks increased (Figure 2).

### 3.6. Block and Urea Consumption on Water Intake and Urine Excretion

No significant difference (*p* > 0.05) was observed between water intake and urine excretion with an increase in the urea level in the blocks (Figure 3). Water intake ranged from 1939 to 2308 mL/day, and urine excretion ranged from 805 to 1016 mL/day. With a higher block intake, increases in water consumption (r = 0.442; *p* = 0.051) and urinary excretion (r = 0.441; *p* = 0.051) were observed. These data also suggest that as block intake increased, fecal excretion increased (r = 0.540; *p* = 0.014) and DM digestibility was reduced (r = −0.525; *p* = 0.018).

## 4. Discussion

Goats are known for their selective feeding behavior since they are not as efficient at utilizing structural carbohydrates as cattle and sheep. Their selectivity is inversely related to their ability to retain and digest fiber in the rumen [3,6]. In this study, doe diet selection was evident, as the NDFom of the forage consumed was lower (ranging from 56.2 to 59.7%), while the NDFom of forage rejected was higher (ranging from 85.4 to 87.0%) than those of the Buffelgrass hay that was offered (69.2%).

When supplementing nutrients for goats consuming low-quality forages in range or confinement conditions, it is crucial to consider the synchronization of nitrogen and carbohydrate degradability in the rumen to optimize rumen fermentation [17]. In this study, DM digestibility (%) was lower than NDFom digestibility (%). With the increased consumption of ash, fecal DM excretion also increased, resulting in lower DM digestibility. Additionally, as block intake increased, more water was consumed, and more urine was excreted, primarily due to the consumption of salt.

Supplementing with CP has been shown to increase the intake and digestibility of hay in cattle, sheep, goats, and other ruminants [23,24,25]. However, in this study, urea supplementation did not affect the intake or digestibility of hay. A review of the literature revealed other studies that have also failed to demonstrate a response in this regard in wethers [26] and steers [27,28,29]. Currier et al. (2004b) [24] suggested that a possible explanation for the lack of a forage intake response could be the NDFom intake, based on the results from Mertens (1985, 1994) [30,31], which showed that DM intake is maximized when NDFom intake is approximately 12.5 g/kg BW/day. This is consistent with other studies indicating that intake is sensitive to forage NDF in small ruminants [30,31,32]. In this study, NDFom intake ranged from 361.8 to 421.5 g/d, which, considering a body weight of 18 kg, would imply an approximate intake of 22 g/kg BW/day.

This study also demonstrated that urea supplementation did not affect NDFom intake or digestibility. These results align with those reported by Chanjula and Ngampongsai (2008) [33], who used native Thailand-Anglo Nubian crossbred doelings with an average weight of 19 kg and fed them elephant grass and yucca-based diets with four urea levels (0, 1, 2, and 3%). The authors found no significant difference (*p* > 0.05) between treatments regarding digestion coefficients for DM, OM, CP, NDF, and ADF. Schacht et al. (1992) [34] did not observe a growth response with the supplementation of urea (5 g/day) or molasses (140 g/day) alone to goat kids grazing native vegetation (Caatinga) in Northeast Brazil. However, when both were supplemented, the average daily gain was doubled, suggesting the need for an energy source to enable urea utilization by rumen bacteria for synthesizing microbial protein.

Other factors, such as DM intake and its effect on rumination, influenced ruminal pH [35]. Urea supplementation of a low protein diet would increase fermentation, resulting in VFA production, which may explain the reduction in ruminal pH [36]. A higher ruminal pH could be attributed to a buffering effect of ammonia nitrogen resulting from urea breakdown in the rumen [37,38]. The significant variation in the ammonia concentration found by Smith et al. (1980) [39] in fibrous diets containing urea was because urea is 100% soluble, rapidly increasing rumen ammonia concentrations after ingestion, as rumen microorganisms may not have sufficient energy available to fully metabolize it.

Diet digestibility and the rate of passage are reduced if the nutrient requirements of rumen bacteria are not met [38]. The nitrogen requirements for maximum ruminal microbial growth are primarily dependent on digestible dry matter intake [39]. The solubility and degradability of dietary protein play a role in protein availability to satisfy the nitrogen needs of microorganisms. Therefore, the required nitrogen level in the rumen to support a maximum rate of feed passage is expected to vary with carbohydrate digestibility in the rumen. The results from various studies with beef cattle, such as NRC (1987) [40], suggest that most diets satisfy the requirement of 6 to 8% CP for normal rumen function. The value of 8% CP that was estimated for adult doelings in this study aligns with values obtained in other studies reported for beef cattle [38] and goats [41].

Adamu et al. (1989) [42] observed that, in animals fed corn stover silage-based diets and supplemented with a protein concentrate containing various urea levels, the maximum microbial growth, measured through bacterial nitrogen reaching the duodenum, occurred when rumen ammonia levels reached 4.9 mg/dL. For animals fed four times a day, the optimum ammonia level to maximize dry matter intake and digestibility was approximately 13.3 mg/dL. In our study, this value was achieved with a block containing 2% urea. These authors concluded that to maintain the rumen ammonia levels in animals fed once per day, a value exceeding 18.2 mg/dL at two hours was required for maximum feed intake.

## 5. Conclusions

Urea supplementation is primarily recommended for goats fed low-quality forages. Buffelgrass generally contains less crude protein than what is required for normal rumen function or for maintaining goats. In this study, increasing the urea content of molasses blocks by up to 6% significantly increased nitrogen intake, retention, and balance in goats. To maintain a positive nitrogen balance, a minimum of 8% crude protein is required in urea-based diets.

## Figures and Tables

**Figure 1 animals-13-03370-f001:**
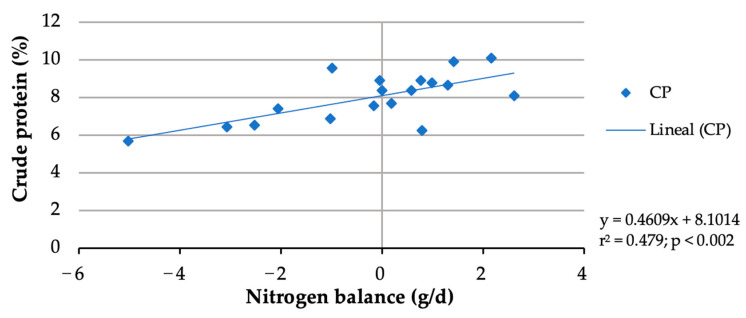
Crude protein requirement estimation for maintenance of doelings consuming Buffelgrass hay and multinutrient blocks with various urea levels.

**Figure 2 animals-13-03370-f002:**
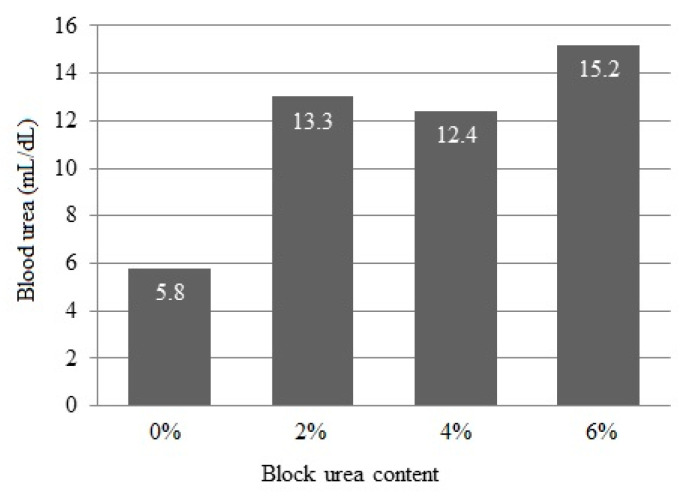
Blood serum urea concentration of doelings consuming Buffelgrass hay and supplemented with multinutrient blocks with various urea levels.

**Figure 3 animals-13-03370-f003:**
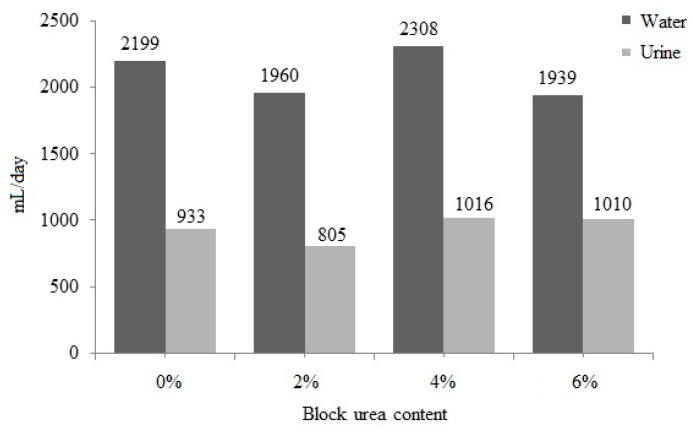
Water intake and urinary excretion of doelings consuming Buffelgrass hay and supplemented molasses blocks with various urea levels.

**Table 1 animals-13-03370-t001:** Ingredient and nutrient composition of molasses blocks with various urea levels.

Item	0	2	4	6
**Ingredient (kg/ton)**				
Molasses	350	350	350	350
Soybean hulls	60	60	60	60
Cracked corn	335	315	295	275
Urea	0	20	40	60
Salt	130	130	130	130
Calcium oxide	100	100	100	100
Mineral-vitamin premix ^1^	25	25	25	25
**Chemical composition**				
Crude protein, %	4.1	10.0	14.6	21.7
Metabolizable energy, Mcal/kg	2.095	2.041	1.987	1.933
Neutral detergent fiber, %	19.9	18.1	14.1	13.4
Ether extract, %	1.0	1.1	1.0	1.0
Ash, %	32.3	34.3	33.6	34.9
Non-fiber carbohydrates, %	42.7	36.5	36.7	29.0

^1^ Mineral-vitamin premix: Cu, 100 mg/kg; Zn, 150 mg/kg; Mn, 50 mg/kg; Se, 1.0 mg/kg; I, 2.5 mg/kg; Co, 0.75 mg/kg; vitamin A, 21,000 UI/kg; vitamin E, 1060 UI/kg.

**Table 2 animals-13-03370-t002:** Effects of urea content of multinutrient blocks supplemented to doelings consuming Buffelgrass hay on dry matter and neutral detergent fiber intakes and digestibility.

	Block Urea Content, %					P ^2^	
Item	0	2	4	6	EE ^1^	L	Q
Body weight							
Initial, kg/d	19.2	18.0	17.5	17.9	2.06	0.509	0.590
Final, kg/d	21.9	19.9	20.8	20.3	2.45	0.622	0.671
Change, kg/d	0.28	0.21	0.37	0.27	0.094	0.706	0.809
Dry matter							
Forage, g/d	571	542	543	554	42.8	0.796	0.645
Block, g/d	165	121	168	122	30.0	0.557	0.977
Total, g/d	736	663	712	676	57.3	0.616	0.742
Fecal DM, g/d	158	143	154	157	20.3	0.911	0.659
DM digestibility, %	79.3	78.6	78.5	77.0	3.28	0.373	0.803
NDFom							
Intake, g/d	421.5	361.8	376.7	368.9	38.2	0.415	0.508
Feces, g/d	82.3	73.0	76.6	81.9	1.95	0.966	0.515
Digestibility, %	80.9	80.0	79.9	78.0	0.71	0.236	0.789

^1^ SEM, standard error of the mean. ^2^ P, probability; L, linear effect; Q, quadratic effect. Metabolizable energy was calculated using values reported for ingredients by the National Research Council (NRC, 2007).

**Table 3 animals-13-03370-t003:** Eating, rumination, and total chewing times of doelings consuming Buffelgrass and supplemented multinutrient blocks with various urea levels.

	Block Urea Content, %					P ^2^	
Variable	0	2	4	6	SEM ^1^	L	Q
Chewing times, min/d							
Eating	424	412	471	447	33.6	0.469	0.878
Rumination	444	340	351	400	27.5	0.346	0.013
Total	868	752	822	847	45.0	0.972	0.136
Ruminal pH	6.62	6.75	6.72	6.58	0.048	0.457	0.012

^1^ SEM, standard error of the mean. ^2^ P, probability; L, linear effect; Q, quadratic effect.

**Table 4 animals-13-03370-t004:** Nitrogen balance of doelings consuming Buffelgrass hay and supplemented multinutrient blocks with various urea levels.

	Urea, %					P ^2^	
Item	0	2	4	6	SEM ^1^	L	Q
Urine, mL/d	1063	805	1016	753	167.7	0.375	0.988
N intake, g/d							
Forage, g/d	6.120	5.994	5.928	6.010	0.453	0.998	0.822
Block, g/d	1.065 ^c^	1.924 ^bc^	3.924 ^ab^	4.600 ^a^	0.571	0.000	0.875
Total	7.180	7.922	9.848	10.605	0.796	0.006	0.993
Fecal N excretion, g/d	7.110	5.772	6.228	6.438	0.867	0.706	0.388
Urinary N excretion, g/d	2.528	2.242	3.100	3.275	0.463	0.174	0.627
N balance, g	−2.458 ^b^	−0.092 ^ab^	0.520 ^ab^	0.895 ^a^	0.762	0.010	0.213
N retention, %	−35.93 ^b^	−0.84 ^ab^	5.73 ^a^	10.06 ^a^	9.22	0.005	0.118

^1^ SEM, standard error of the mean. ^2^ P, probability; L, lineal effect; Q, quadratic effect. Different letters indicate significant difference (*p* < 0.05).

## Data Availability

The data presented in this study are available on request from the corresponding author.

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
