# Peer review of "Nitrogen Utilization in Goats Consuming Buffelgrass Hay and Molasses-Based Blocks with Incremental Urea Levels"

_animals, 2023, doi:10.3390/ani13213370_

Round 1

Reviewer 1 Report

Comments and Suggestions for Authors

This paper (Nitrogen utilization in goats consuming Buffelgrass hay and molasses-based blocks with various urea levels.) aims “to investigate nitrogen utilization in goats fed low-quality hay and supplemented molasses blocks with urea”. The topic of the study falls within the general scope of the journal.

However, there are some questionable points that need to be carefully considered.

Starting from the section Mat. & Met., the authors did not report the chemical characteristic of the forage used in the experiment, neither of the administered hay nor of the orts, although in the manuscript it is reported they were analyzed (line 108). Only the crude protein concentration is reported. This lack is very important because the hay represented the most part of the total dry matter of the ingested diets. The lack of this information makes difficult for the reader to have an idea of the quality of the forage. In this view, looking to the results reported in table 2 concerning the diet NDF digestibility, the values of this parameter (range of 86.9-87.6%) seem to be unreliable because too high. In the manuscript the authors did not make any comment regarding this.

Moreover, another lack is that the nitrogen digestibility values of the different diets are not reported. However, looking to the data reported in table 4, it is possible to calculate them. In the case of the diet with block without urea (treatment 0), the calculated N digestibility is lower than 1%, ((7.180-7.11)/7.18x100) that seems an unreliable value, considering that the molasses block without urea, based on its composition reported in table 1, should have a N digestibility of about 60%. Because the N intake from the block without urea was about 15% (1.065/7.180) of total N intake, a value of about 1% in N digestibility of the entire diet appears underestimated, also if the N digestibility of the hay were 0. The authors did not make any comment regarding this, also in this case. As a consequence, this underestimation reflects on the N balance, questioning on the accuracy of the regression equation reported in figure 2.

In relation to the above observations, this manuscript appears not to be robust enough to be published in its current form.

Lastly, it would be advisable to report in the manuscript the data relating to the weight gain of the animals. 

Other minor observations are reported below.

Table 2: how did the authors determine or calculate the ME concentration of the molasses-blocks?

Table 2: the ether extract concentrations seem to be expressed in g/kg and not in %.

Line 93: the concentration of vitamins of the mineral-vitamin premix should be reported.

Line 121: the authors did not mention about the possible partial loss of N by volatilization due to drying the fecal samples in oven at 55°C.

Line 124: the authors did not mention how it was avoided the loss of N from the urine from the plastic containers.

Table 3: the authors did not mention how did they collect the rumen fluid. In the manuscript it is not reported.

Author Response

Dear Reviewer,

Thanks for considering this manuscript for evaluation by referees and for the valuable comments of the reviewers. We revised the manuscript according to comments and prepared a point-by-point response. I believe the revised manuscript was improved and I hope these changes are satisfactory.

Reviewer 1

  1. Materials and Methods: Starting from the section Mat. & Met., the authors did not report the chemical characteristics of the forage used in the experiment, neither of the administration of hay nor the orts, although in the manuscript it is reported they were analyzed (line 108). Only the crude protein concentration is reported. The lack is very important because the hay represented the most part of the total dry matter of the ingested diets. The lack of this information makes difficult for the reader to have an idea of the quality of the forage.

Answer: Thank you for your observation. Certainly, we should have reported this information. We are adding it to the materials and methods section.

…… and Buffelgrass (Cenchrus ciliaris) hay ad libitum. Chemical composition of Buffel grass was, crude protein, 6.47%; crude fat, 0.94%; NDF, 69.2%; ADF, 48.1%; ADL, 7.79%; and ash, 9.36%.

  1. In this view, looking to the results reported in table 2 concerning the diet NDF digestibility, the values of this parameter (range of 86.9 to 87.6%) seem to be unreliable because too high. In the manuscript the authors did not make any comment regarding this.

Answer: Molasses-urea blocks had a high ash content, varying from 32.3 to 34.9% because of the inclusion of large quantities of calcium oxide, salt, and other mineral sources. For this reason, energy density of the blocks was low (1.933 to 2.095 Mcal/kg). Due to the high inorganic matter content of blocks, DM digestibility was relatively low in all treatments and lower than NDFom digestibility. Although NDFom contented in Buffelgrass was relatively high (69.2%), due to doe diet selection, forage consumed contained less NDFom (52.9, 50.8, 49.4, and 52.1% for 0, 2, 4, and 6% urea treatments, respectively). NDFom in orts were 87.0, 86.5, 85.5, and 85.4%, respectively.

  1. Moreover, another lack is that the nitrogen digestibility values of the different diets are not reported. However, looking to the data reported in table 4, it is possible to calculate them.

Answer: We can calculate the apparent N digestibility values of the different diets; however, these would not be true digestibility values. A fraction of the fecal N should be fecal metabolic nitrogen. Since most of the crude protein of the diets is from non-protein nitrogen (urea), we believe that a digestibility value would not be appropriate. This is the reason why we focused on N retention and balance.

  1. In the case of the diet with block without urea (treatment 0), the calculated N digestibility is lower than 1%, (7.180-7.11/7.11x100) that seems an unreliable value, considering that molasses block without urea, based on its composition reported in table 1, should have a N digestibility of about 60%. The authors did not make any comments regarding this, also in this case. Consequently, this underestimation reflects on the N balance, questioning on the accuracy of the regression equation reported in figure 2.

Answer: With the diets containing 0 % urea, total nitrogen intake was 7.180 g/d and fecal N excretion was 7.110 g/d; the later should include FMN excreted in the feces. The digestibility value was underestimated by the FMN. In Figure 2, N retention and balance considered urinary N excretion. What we wanted was to determine the minimum crude protein (%) in the diet for a positive N balance.

  1. Lastly, it would be advisable to report in the manuscript the data relating to the weight gain of the animals.

Answer: Thank you, we understand your suggestion, but we used adult does, so body weight change would be more appropriate. In Table 2 we included the initial and final weights, and body weight change of the doelings during the 9-day sampling period.

Other minor observations are reported below.

  1. Table 2: how did the authors determine or calculate the ME concentration of the molasses-blocks?

Answer: Metabolizable energy was calculated using values reported for ingredients by the National Research Council (NRC, 2007).

National Research Council. 2007. Nutrient Requirements of Small Ruminants: sheep, goats, cervids, and new world camelids.

  1. Table 2: the ether extract concentrations seem to be expressed in g/kg and not in %.

Answer: Thank for your observation, we changed in the manuscript.

  1. Line 93: the concentration of vitamins of the mineral-vitamin premix should be reported.

Answer: Thank for your observation, we added in the manuscript.

  1. Line 121: the authors did not mention about the possible partial loss of N by volatilization due to drying the fecal samples in oven at 55°C.

Answer: At 55°C in an air-draft oven, nitrogen was not lost (volatilize) from the supplements in our study. The percentage of urea apparently lost as moisture, increases exponentially as oven temperature increases up to 130°C. The following paper evaluated urea loss as moisture in various feeds.

“Thiex, N., Richardson, C.R. Challenges in measuring moisture content of feeds. 2003. J. Anim. Sci.12, 3255-66”.

  1. Line 124: the authors did not mention how it was avoided the loss of N from the urine from the plastic containers.

Answer: Urine excretion was measured daily, and a representative sample was immediately frozen to avoid N loss. It is difficult to measure the nitrogen loss from urine in metabolic cages, but we assume it should be small.

  1. Table 3: the authors did not mention how did they collect the rumen fluid. In the manuscript it is not reported.

Answer: Thank for your observation, we added in the manuscript methodology.

Rumen fluid collection and pH analysis

To obtain the rumen fluid samples from the does, an esophageal probe was inserted orally. The rumen fluid was collected in 50 mL conical tubes and pH was immediately measured with a Beckman pH meter.

Reviewer 2 Report

Comments and Suggestions for Authors

Language is difficult to understand. A very preliminary study and a lot of work has been done on this topic. I think data is not sufficient for publication in such a high-quality Journal.

Line 17: in ruminants or of ruminants? In the sentence ‘Protein in the diet is vital for growth and health in ruminant………..’

Line 18-19: sentence is not clear ‘. However, protein is also one of the most expensive components in animals’ diets and good  protein sources are sometimes difficult to find….’

Line 20=21: sentence is not clear ‘In the rumen, urea is transformed by microorganisms which in turn can produce protein that can be assimilated by the animal host’ for example transformed into? What kind of protein produced? What do you mean by animal host

Line 27: could you state this statement for monogastric ? ‘Urea is an inexpensive nonprotein nitrogen source commonly used in animal nutrition..’

Line 35: check abbreviation is correct? ‘or neutral detergent fiber intake (NDF)…’

No p values in abstract

No conclusion in abstract

Can you justify 9 days data for publication of results/9 days data is not reliable

I will review the manuscript if the language of the manuscript is sufficiently improved and the authors provided answer of the above comments  

Comments on the Quality of English Language

very poor 

Author Response

Dear Reviewer,

Thanks for considering this manuscript for evaluation by referees and for the valuable comments of the reviewers. We revised the manuscript according to comments and prepared a point-by-point response. I believe the revised manuscript was improved and I hope these changes are satisfactory.

Reviewer 2

  1. Line 17: in ruminants or of ruminants? In the sentence ‘Protein in the diet is vital for growth and health in ruminant...........’

Answer: Protein in the diet is vital for growth and health “of” ruminants and other animal

  1. Line 18-19: sentence is not clear ‘. However, protein is also one of the most expensive components in animals’ diets and good protein sources are sometimes difficult to find....’

Answer: Protein is also one of the most expensive components in the animals’ diet and good protein sources are sometimes difficult to find….

  1. Line 20=21: sentence is not clear ‘In the rumen, urea is transformed by microorganisms which in turn can produce protein that can be assimilated by the animal host’ for example transformed into? What kind of protein produced? What do you mean by animal host

Answer: In the rumen, urea is transformed by microorganism into microbial protein that can be assimilated by the animal.

  1. Line 27: could you state this statement for monogastric ? ‘Urea is an inexpensive nonprotein nitrogen source commonly used in animal nutrition..’

Answer: Urea is an inexpensive nonprotein nitrogen source commonly used in ruminant nutrition

  1. Line 35: check abbreviation is correct? ‘or neutral detergent fiber intake (NDF)...’

Answer: did not affect dry matter (DM) or neutral detergent fiber (NDFom) intake or digestibility.

  1. No p values in abstract No conclusion in abstract.

Answer: we appreciate your suggestion, but in many scientific articles including p values in the abstract is acceptable. About the conclusions, we agree with you in that p values shouldn’t be included.

  1. Can you justify 9 days data for publication of results/9 days data is not reliable.

Answer:  Thank you for your observation, but for digestion and metabolism trails, it is common to use a period of 7 days or more for feces and urine sample collection.

Reviewer 3 Report

Comments and Suggestions for Authors

Line 85 – It's not clear for how many days the total collections of leftovers, feces, and urine were made. In line 85, it is mentioned that the collection was carried out over 9 days, while in line 123, it states a collection over 7 days. It would be helpful to clarify the duration of the collection period more explicitly and specify the types of collections conducted.

Line 92 – Table 1 – The unit of measurement used to present the data in the table needs to be identified.

Line 138 – The statistical description is confusing. At one point, orthogonal linear and quadratic contrasts are mentioned, followed by average testing. Regression would be more appropriate to understand the relationship between levels of urea inclusion on a continuous scale and would enhance the presentation of the study's results.

Line 216 – The first paragraph of the discussion is confusing and repetitive, as it restates information that should be confined to the introduction. Additionally, the hypothesis is reiterated, which is unnecessary for this section.

Author Response

Dear Reviewer,

Thanks for considering this manuscript for evaluation by referees and for the valuable comments of the reviewers. We revised the manuscript according to comments and prepared a point-by-point response. I believe the revised manuscript was improved and I hope these changes are satisfactory.

Reviewer 3

1. Line 85 – It's not clear for how many days the total collections of leftovers, feces, and urine were made. In line 85, it is mentioned that the collection was carried out over 9 days, while in line 123, it states a collection over 7 days. It would be helpful to clarify the duration of the collection period more explicitly and specify the types of collections conducted.

Answer: Your observation is correct; we made a mistake in line 122; it should be 9 days.

2. Line 92 – Table 1 – The unit of measurement used to present the data in the table needs to be identified.

Answer: we added the unit of measurement in the manuscript (kg/ton).

3. Line 138 – The statistical description is confusing. At one point, orthogonal linear and quadratic contrasts are mentioned, followed by average testing. Regression would be more appropriate to understand the relationship between levels of urea inclusion on a continuous scale and would enhance the presentation of the study's results.

Answer: Thank you for your comment. We used polynomial contrast since we detected nonlinear relationships between variables. We were interest in determining linear or quadratic effects of the urea levels on dependent variables such as nitrogen balance, intake, and digestion of nutrients. Additionally, we presented a regression between nitrogen balance (g/d) and diet crude protein concentration (%) to determined crude protein requirement in the diet of doelings at zero-nitrogen balanced. Correlations were also presented for some variables.

4. Line 216 – The first paragraph of the discussion is confusing and repetitive, as it restates information that should be confined to the introduction. Additionally, the hypothesis is reiterated, which is unnecessary for this section.

Answer: Thank for your comment, we deleted the first paragraph.

Reviewer 4 Report

Comments and Suggestions for Authors

The manuscript ID animals-2562522 describes the nitrogen utilization in Anglo Nubian crossbred goats consuming Buffelgrass hay and molasses-based blocks with various urea levels. This topic is research related to protein nutrition in goats fed low-protein roughage. In contrast to previously published information, this paper is characterized by the study of Anglo-Nubian hybrid goats consuming Buffelgrass hay. The materials and methods for this experiment are fine. The conclusions are consistent with the evidence and arguments presented. References are good.

I recommend that you revise your manuscript referring to the following.

There are already many reports on the effects of urea supplementation on ruminants fed low-protein roughage. One of a feature of this paper is the use of Anglo-Nubian hybrid goats consuming Buffelgrass hay. In the Introduction, could you describe the characteristics of Buffelgrass and describe for the first time the effect of urea addition on this hay-consuming Anglo-Nubian crossbred goat?

This paper did not have Figure 1.

L176 p=0.001 => p<0.001

or Table4 Linear effect probability of N intake (Block) 0.000 => 0.001

L207-210 Should the author describe the correlation results? If it is necessary to describe the results of the correlation, please also show them in the figure. Please provide further implications of the correlation results in the Discussion.

Author Response

Dear Reviewer,

Thanks for considering this manuscript for evaluation by referees and for the valuable comments of the reviewers. We revised the manuscript according to comments and prepared a point-by-point response. I believe the revised manuscript was improved and I hope these changes are satisfactory.

Reviewer 4

  1. There are already many reports on the effects of urea supplementation on ruminants fed low-protein roughage. One of a feature of this paper is the use of Anglo-Nubian hybrid goats consuming Buffelgrass hay. In the Introduction, could you describe the characteristics of Buffelgrass and describe for the first time the effect of urea addition on this hay- consuming Anglo-Nubian crossbred goat?

Answer: Buffelgrass (Cenchrus ciliaris) is the predominant warm season perennial grass of northeastern Mexico, and common in other semi-arid regions of the world. It is highly productive and tolerant to the periodic droughts that occur in these region [15]. The objective of this study was to explore the effect of molasses blocks with various levels of urea on nitrogen utilization in Anglo-Nubian female doelings.

R.G. Ramírez, J. Huerta, J.R. Kawas, D.S. Alonso, E. Mireles, M.V. Gómez. Performance of lambs grazing in a buffelgrass (Cenchrus ciliaris) pasture and estimation of their maintenance and energy requirements for growth. Small Ruminant Research. 1995, 17, 117-121.

  1. This paper did not have Figure 1.

Answer: we have changed the number of the figures.

  1. L176 p=0.001 => p<0.001

Answer: yes, it is p<0.001.

or Table 4 Linear effect probability of N intake (Block) 0.000 => 0.001

Answer: yes, it is p<0.001.

  1. L207-210 Should the author describe the correlation results? If it is necessary to describe the results of the correlation, please also show them in the figure. Please provide further implications of the correlation results in the Discussion.

Answer: Nutrient supplementation for goats consuming low quality forages in range or confinement conditions should consider the synchronization of nitrogen and carbohydrate degradability in the rumen for optimum rumen fermentation [17]. In this study, DM digestibility (%) was lower than NDFom digestibility (%); with more ash consumed, fecal DM excretion increased, resulting in a lower DM digestibility. Also, more water was consumed, and more urine was excreted as block intake increased, primarily attributing these increments to the consumption of salt.

Round 2

Reviewer 1 Report

Comments and Suggestions for Authors

Comments to the revised manuscript

Title: Nitrogen utilization in goats consuming Buffelgrass hay and molasses-based blocks with various urea levels.

The authors revised the manuscript following the suggestions of the reviewers. However, there are some points that need to be improved again in order to let the manuscript more clear and readable.

- Line 137: the AU indicated that the goats did diet selection. However, it is not indicated how much it was. It is important to quantify this amount for a better understanding of diet NDF digestibility. Considering the new data reported for NDF of the administered forage, for the NDF of the orts and of the ingested forage, it seems that the consumption of the Buffelgrass was on average about 61% of the offered forage. So, please, report the amount (in percentage) of the refusals, and add also the ADL concentration determined in the orts. NDF is not enough.

- Table 2: please, report in the footnote of the table how you calculate the ME values, as you wrote in your reply.

- Lines 232-233 (“since they are not…and sheep”): what do you mean with efficient utilizers? You mean that goats digest at less extent the fiber? Please, add references to support this.

- 2.3 paragraph: to avoid N losses from the feces it should be better to determine the N concentration on wet feces. To avoid N losses from the urine it should be better to add a solution of 20% of sulfuric acid in the vessel for urine collection in order to reach a pH of about 2 at the end of the day in the acidified urine. So, in the manuscript the AU should at least underline that their data in some way overestimated the N retention. These considerations are at least in part referred to the paper of Spanghero and Kowalski (J. Dairy Sci. 104:7725–7737 https://doi.org/10.3168/jds.2020-19656).

Author Response

Dear Reviewer

We revised the manuscript according to comments and prepared a point-by-point response. I believe the revised manuscript was improved and I hope these changes are satisfactory.

Reviewer 1

     The authors revised the manuscript following the suggestions of the reviewers. However, there are some points that need to be improved again in order to let the manuscript more clear and readable.

- Line 137: the AU indicated that the goats did diet selection. However, it is not indicated how much it was. It is important to quantify this amount for a better understanding of diet NDF digestibility. Considering the new data reported for NDF of the administered forage, for the NDF of the orts and of the ingested forage, it seems that the consumption of the Buffelgrass was on average about 61% of the offered forage. So, please, report the amount (in percentage) of the refusals, and add also the ADL concentration determined in the orts. NDF is not enough.

Answer: We understand observation, unfortunately we did not analyze ADL. But we included the following information:

  • Thank you for your observation about the percentage of orts. As you mentioned, certainly there were high amounts of forage rejected.
  • Hay rejected was high, the percent of orts being 30.5, 34.5, 33.9, and 32.6% of Buffelgrass offered, for 0, 2, 4, 6% of urea treatments, respectively.
  • Fecal NDFom was 52.9%, 50.8%, 49.4%, and 52.1% for 0, 2, 4, and 6% urea, respectively. These low fecal NDFom values suggest that forage consumed was low in NDFom content.

- Table 2: please, report in the footnote of the table how you calculate the ME values, as you wrote in your reply.

Answer: We added in the footnote of Table 2.

- Lines 232-233 (“since they are not...and sheep”): what do you mean with efficient utilizers? You mean that goats digest at less extent the fiber? Please, add references to support this.

Answer: Goats are known for their selective feeding behavior since they are not as efficient utilizing structural carbohydrates as are cattle and sheep. Their selectivity is inversely related to their ability to retain and digest fiber in the rumen [3, 6]. In this study, doe diet selection was evident, as the NDFom of the forage consumed was lower (ranging from 56.2 to 59.7%), while the NDFom of forage rejected was higher (ranging from 85.4 to 87.0%) than that of the Buffelgrass hay that was offered (69.2%). Also, fecal NDFom values were low.

- 2.3 paragraph: to avoid N losses from the feces it should be better to determine the N concentration on wet feces. To avoid N losses from the urine it should be better to add a solution of 20% of sulfuric acid in the vessel for urine collection in order to reach a pH of about 2 at the end of the day in the acidified urine. So, in the manuscript the AU should at least underline that their data in some way overestimated the N retention. These considerations are at least in part referred to the paper of Spanghero and Kowalski (J. Dairy Sci. 104:7725–7737 https://doi.org/10.3168/jds.2020-19656).

Answer: Thank you for your recommendation, unfortunately we did not analyzed nitrogen in wet feces. However, feces were dried at a low temperature (55 °C).

Reviewer 2 Report

Comments and Suggestions for Authors

Dear Editor

The author's conclusions do not reflect the actual study results. The data of the current study is not sufficient to publish in a high-quality journal.

Dear Editor

I rejected the manuscript in the previous revision because the author did not provide the details of why they were doing this research. Please check the introduction of this study, which does not reflect the need for study and a clear study plan supported by previous literature. Rather authors provided ambiguous statements  in the introduction section and misleading information in the discussion. The introduction lacks the need for study, hypothesis, and clear objective.

Review comments

Line 18-19: ‘……………….and good protein sources are sometimes difficult to find’ sentence is incomplete and did not provide sufficient information

Line 23-25: statement did nothing new, already a a lot of work has been done ‘The findings and discussion in this paper contribute to a better understanding of nitrogen utilization in goats using urea as a non-protein nitrogen source’

Line 25-26: ‘The use of goats to produce meat and milk is challenged by environmental and nutritional factors’

Did you study any environmental factor? What is the significance of this sentence here

Line 27-28: do you think, this feeding regime could provide all required nutrient for growth, maintenance and production to the goat? If no, how can these results be justified ‘The aim of this study was to investigate nitrogen utilization in goats fed low-quality hay and supplemented molasses blocks with urea’

Line 34-36: these are results or what ‘The minimum CP requirement of 8% for maintenance in doelings consuming low quality forage with a urea-based supplement was determined by regression analysis between CP intake (% of DM) and N balance (r2 = 0.479; p < 0.002)’ I am unable to understand

Line 37-38: have no meaning ‘The results contribute to a better understanding of nitrogen utilization in goats fed low-quality hay with urea supplementation’ conclusion should be based on what the actual results are

Line 43: what do you mean by ‘other useful products’

Line 44: ‘……………thrive in environmentally adverse conditions’ reference missing and information is misleading

Line : ‘continue to grow’ what does it represent ?

Line 46-47: meaning are not clear ‘………….goats make an important contribution to local economies and smallholder livelihoods’

Line 47-49: anything related to your study? These sentence are not clear ‘However, the potential of goats to grow, reproduce, and generate useful products is challenged by environmental (e.g., weather) and nutritional (e.g., diet  composition) factors’

Line 52: ‘This is a concern’ what does it reflect

Line 52-53: sentence structure is not clear ‘. This is a concern because underfed animals cannot display their full poten- 52 tial to maintain health, reproduce, and at the same time produce meat and milk’

Line 53-56: ‘Nutrient supplementation to animals consuming low-quality forages can have positive influence but should be specific and consider protein and mineral concentrations of forages in different regions’ I am not sure what does nutrient supplement mean? Why author giving blind statements

Line 57-59: not clear? In animals? In plants? Where? ‘Dietary protein and nitrogen supply are vital for health because amino acids and nitrogen are involved with the production of essential molecules like antibodies, enzymes, and neurotransmitters’ please also check the quality

Line 59-61: strange statement ‘However, nitrogen metabolism is not well understood because it involves different routes for absorption and excretion depending on the type and quantity of dietary protein and protein synthesis rates in different organs’ please check the reference

Line 78: what does ‘This study was accepted by the Joint Graduate Program….’ Accepted mean ?

Line 106: what do you mean by ‘registered’

Line 107: ‘dedicated’…?

Line 111: ‘Offered and rejected hay samples were frozen for further analysis’…..?

Statistical analysis
Poor description of how the data were analyzed.
What were the fixed and random effects used in the model? What procedure was used?
Repeated measured? If so, how appropriate covariance structure was chosen?
Why were contrast statements not used to test for linear and quadratic effects?
LSMEANS should be tested at <0.05 only. Extremely significantly?
line 232-237: misleading information ‘Goats are known for their selective feeding behavior since they are not as efficient 232 utilizers of structural carbohydrates as are cattle and sheep; their selectivity is inversely 233 related to their ability to retain and digest fiber in the rumen [3, 6]. In this study, doe diet 234 selection was noticeable since NDFom of forage consumed was less (56.2 to 59.7%) and 235 NDFom of forage rejected was more (85.4 to 87.0%) than that of the Buffelgrass hay that 236 was offered (69.2%)’ you can not compare your study with them because they are in well fed condition while your study doesn’t meet the criteria to fulfill the all requirement of the animals

Line 245: misleading information of reference ‘CP supplementation has been shown to increase the intake and…’ studies representing degradable or undegradable protein not CP

Line 247: ‘A revision of the literature…’ …………..?

Comments on the Quality of English Language

poor

Author Response

Dear Reviewer

We revised the manuscript according to comments and prepared a point-by-point response. I believe the revised manuscript was improved and I hope these changes are satisfactory.

Reviewer 2

Line 18-19: ‘...................and good protein sources are sometimes difficult to find’ sentence is incomplete and did not provide sufficient information

Answer: Protein is also one of the most expensive components in the animals' diet, and finding good protein sources can sometimes be challenging.

Line 23-25: statement did nothing new, already a lot of work has been done ‘The findings and discussion in this paper contribute to a better understanding of nitrogen utilization in goats using urea as a non-protein nitrogen source’

Answer: The findings and discussions in this paper contribute to a better understanding of nitrogen utilization in goats, using urea as the main nitrogen source.

Line 25-26: ‘The use of goats to produce meat and milk is challenged by environmental and nutritional factors’. Did you study any environmental factor? What is the significance of this sentence here

Answer: The use of goats for meat production faces challenges from environmental and nutritional factors, primarily in underdeveloped countries. Most of the goat meat production is being held in poor environmental adverse conditions.

Line 27-28: do you think, this feeding regime could provide all required nutrient for growth, maintenance, and production to the goat? If no, how can these results be justified ‘The aim of this study was to investigate nitrogen utilization in goats fed low-quality hay and supplemented molasses blocks with urea’

Answer: Under the conditions of this study, this feeding regime (Buffelgrass hay and molasses-urea blocks with mineral-vitamin A premix) can provide nutrients for maintenance and moderate production of the meat goat.

Line 34-36: these are results or what ‘The minimum CP requirement of 8% for maintenance in doelings consuming low quality forage with a urea-based supplement was determined by regression analysis between CP intake (% of DM) and N balance (r2 = 0.479; p < 0.002)’ I am unable to understand

Answer: Using a regression equation, we estimated the minimum crude protein content required in the diet to attain a positive nitrogen balance. We believe that this crude protein level is adequate to maximize rumen fermentation, using urea as the only nitrogen source, in addition to the small quantity of crude protein provided by some ingredients in the blocks. We used molasses as a source of fermentable carbohydrates and urea as a source of readily available nitrogen in the rumen. However, in this case, we only used urea to gradually increase crude protein in the blocks. The value of 8% of crude protein obtained in this study is similar to several previous studies reported in the literature, but in this case, the increments in crude protein came exclusively from urea.

Line 37-38: have no meaning ‘The results contribute to a better understanding of nitrogen utilization in goats fed low-quality hay with urea supplementation’ conclusion should be based on what the actual results are

Answer: In this study, increasing the urea content of molasses blocks up to 6% significantly in-creased nitrogen intake, retention, and balance in goats.

Line 43: what do you mean by ‘other useful products’

Answer: Goats (Capra aegagrus hircus) are small ruminants capable of producing meat, milk, and other valuable products (cashmere and mohair) for humans.

Line 44: ‘...............thrive in environmentally adverse conditions’ reference missing and information is misleading

Answer: The reference was added. Alexandre, G.; Mandonnet, N. Goat meat production in harsh environments. Small Rumin. Res. 2005, 60, 53-66.

Line 45: ‘continue to grow’ what does it represent?

Answer: The global goat population exceeds one billion and continues to grow, particularly in Asia and Africa [1].

Line 46-47: meaning are not clear ‘.............goats make an important contribution to local economies and smallholder livelihoods’

Answer: In some countries, especially those in semiarid regions, goats play a significant role in local economies and the livelihoods of smallholders [2].

Line 47-49: anything related to your study? This sentence are not clear ‘However, the potential of goats to grow, reproduce, and generate useful products is challenged by environmental (e.g., weather) and nutritional (e.g., diet composition) factors’

Answer: However, the potential of goats to grow, reproduce, and yield useful products (cashmere and mohair) is challenged by environmental factors (e.g., weather) and nutritional factors (e.g., diet composition) [3].

Line 52: ‘This is a concern’ what does it reflect

Answer: This is a concern because undernourished animals cannot fully realize their full potential to maintain health, reproduce, and simultaneously produce meat and milk [4].

Line 52-53: sentence structure is not clear ‘. This is a concern because underfed animals cannot display their full potential to maintain health, reproduce, and at the same time produce meat and milk’

Answer: This is a concern because undernourished animals cannot fully realize their full potential to maintain health, reproduce, and simultaneously produce meat and milk [4].

Line 53-56: ‘Nutrient supplementation to animals consuming low-quality forages can have positive influence but should be specific and consider protein and mineral concentrations of forages in different regions’ I am not sure what does nutrient supplement mean? Why author giving blind statements

Answer: Nutrient supplementation for animals consuming low-quality forages can have a positive impact, but it should be specific and take into consideration the protein and mineral concentrations of forages in different regions [5, 6].

A feed supplement provides nutrients that are deficient in the animals´ diet.

Line 57-59: not clear? In animals? In plants? Where? ‘Dietary protein and nitrogen supply are vital for health because amino acids and nitrogen are involved with the production of essential molecules like antibodies, enzymes, and neurotransmitters’ please also check the quality

Answer: Dietary crude protein and nitrogen supply to the animal are vital for maintaining health, as amino acids and nitrogen are involved in the production of essential molecules such as antibodies, enzymes, and neurotransmitters [7].

Line 59-61: strange statement ‘However, nitrogen metabolism is not well understood because it involves different routes for absorption and excretion depending on the type and quantity of dietary protein and protein synthesis rates in different organs’ please check the reference

Answer: However, nitrogen metabolism is not well understood because it involves different pathways for absorption and excretion, depending on the type and quantity of dietary protein, as well as protein synthesis rates in different organs [8,9].

Line 78: what does ‘This study was accepted by the Joint Graduate Program....’ Accepted mean ?

Answer: This study was approved by the Joint Graduate Program of the Faculties of Agronomy and Veterinary Science of the University of Nuevo León and registered under the code 36397-001290684 in the masters' exam certificate and code 4768 in the digital collection of the masters' degree thesis. 

Line 106: what do you mean by ‘registered’

Answer: Rejected Buffelgrass hay was weighted and recorded in the morning to calculate daily hay intake.

Line 107: ‘dedicated’...?

Answer: At the end of the experimental period, time dedicated to eating, ruminating, or engaging in other activities was recorded every 5 min over a 24-h period [18].

I could not find a better synonym for dedicated.

Line 111: ‘Offered and rejected hay samples were frozen for further analysis’.....?

Answer: Yes, we froze both offered and rejected hay samples until we had an opportunity to analyze them.

Statistical analysis

Poor description of how the data were analyzed. What were the fixed and random effects used in the model? What procedure was used?

Answer: We agree with your observation. We made some corrections to the Statistical Analyses paragraph. The model is as follows:

y i j = m + Ti + S i j

Where:

y i j: The response variable for the ith treatment (where i = 1, 2, 3, 4) and the jth observation treatment group (where j = 1, 2,….., ni).

m: The overall population mean (the average response across all treatments).

Ti: The effect of the ith treatment level (incremental urea level), which represents the difference between the mean response for treatment I and the overall population mean.

The treatments (urea levels) are fixed effect which represent the main factor being study.

2.6. Statistical analysis

All data were analyzed using an analysis of variance for a completely randomized de-sign using Statistics 9 Analytical Software (Tallahassee, FL, United States). The model included treatments, and all possible interactions, with the animal as experimental unit. Animal and the error term were considered random in the model. All variables were analyzed for lineal and quadratic responses to molasses levels using orthogonal contrasts. The Tukey multiple comparison test was used to determine differences among means. Mean p-values were considered statistically significant at p < 0.05. The initial weight of the doelings was considered as a covariate. Correlations coefficients were obtained between block intake, water consumption, urine excretion, fecal excretion, and dry matter digestibility. A lineal regression analysis was performed to determine the maintenance crude protein requirement of adult doelings using diet crude protein and nitrogen balance data.

Repeated measured? If so, how appropriate covariance structure was chosen?

Answer: We did not use a repeated measures design. We took fecal samples during 9 continuous days, frozen them as they were sampled, and at the end of the study, these were dried and ground. The same procedure was followed for the urine samples. Finally, composite samples were obtained for each experimental unit (doelings) for analyses.

Why were contrast statements not used to test for linear and quadratic effects?

Answer: Yes, polynomial contrasts were used to determine linear and quadratic effects in response to incremental urea levels in molasses blocks [14].

LSMEANS should be tested at <0.05 only. Extremely significantly?

Answer: The Tukey multiple comparison test was used to determine differences among means. Mean p-values were considered statistically significant at p < 0.05.

Line 232-237: misleading information ‘Goats are known for their selective feeding behavior since they are not as efficient utilizers of structural carbohydrates as are cattle and sheep; their selectivity is inversely related to their ability to retain and digest fiber in the rumen [3, 6]. In this study, doe diet selection was noticeable since NDFom of forage consumed was less (56.2 to 59.7%) and NDFom of forage rejected was more (85.4 to 87.0%) than that of the Buffelgrass hay that was offered (69.2%)’ you can not compare your study with them because they are in well fed condition while your study doesn’t meet the criteria to fulfill the all requirement of the animals.

Answer: Goats are known for their selective feeding behavior since they are not as efficient utilizing structural carbohydrates as are cattle and sheep. Their selectivity is inversely related to their ability to retain and digest fiber in the rumen [3, 6]. In this study, doe diet selection was evident, as the NDFom of the forage consumed was lower (ranging from 56.2 to 59.7%), while the NDFom of forage rejected was higher (ranging from 85.4 to 87.0%) than that of the Buffelgrass hay that was offered (69.2%).

Line 245: misleading information of reference ‘CP supplementation has been shown to increase the intake and...’ studies representing degradable or undegradable protein not CP

Answer: Supplementing with CP has been shown to increase the intake and digestibility of hay in cattle, sheep, goats, and other ruminants [23, 24, 25]. However, in this study, urea supplementation did not affect the intake or digestibility of hay. A review of the literature revealed other studies that have also failed to demonstrate a response in this regard in wethers [26] and steers [27, 28, 29].

Line 247: ‘A revision of the literature...’ ..............?

Answer: A review of the literature revealed other studies that have also failed to demonstrate a response in this regard in wethers [26] and steers [27, 28, 29].

Reviewer 3 Report

Comments and Suggestions for Authors

I appreciate your revisions to the article in accordance with the suggestions I provided. After a thorough review of the modifications made, I am pleased with the corrections and consider the article ready to proceed with the final review process.

Author Response

Dear Reviewer, thank for your evaluation and comments.